# The promise and challenge of cancer microbiome research

Sumeed Syed Manzoor, Annemiek Doedens and Michael B. Burns*

* Correspondence: mburns16@luc.edu
Department of Biology, Loyola University Chicago, Chicago, IL 60660, USA

## Abstract

Many microbial agents have been implicated as contributors to cancer genesis and development, and the search to identify and characterize new cancer-related organisms is ongoing. Modern developments in methodologies, especially culture-independent approaches, have accelerated and driven this research. Recent work has shed light on the multifaceted role that the community of organisms in and on the human body plays in cancer onset, development, detection, treatment, and outcome. Much remains to be discovered, however, as methodological variation and functional testing of statistical correlations need to be addressed for the field to advance.

## Introduction

The human microbiome is a significant community, with an estimated ratio of one microbial cell per human cell [1] and nearly 500-fold more microbial genes than host genes [2]. This community is dynamically shaped alongside human development from birth through adolescence. It has coevolved with humans to the degree that it plays an integral role in normal, healthy human functioning [3]. Experiments generating and assessing gnotobiotic and germ-free (GF) mice suggest that, while not a requisite component of physiology, this "hidden organ" provides critical functions that allow for normal metabolic and immune functioning [4]. The role of microbiota as a key functional regulator of metabolic homeostasis [5, 6], drug detoxification and metabolism [7–9], and metabolite biosynthesis [10] has been established only recently, and new findings are emerging on a regular basis. Just as microbiota are important in healthy functioning, it has a hand in dysfunction and disorder. Microbial dysbiosis may be loosely described as a human microbiome that does not fulfill all the necessary functions required for health. It has been implicated in metabolic disorders, obesity [5, 11], and immune development, as well as a wide array of disease states [12, 13]. While microbial communities are functionally similar between individuals, they can be wildly dissimilar phylogenetically, a phenomenon that presents unique challenges in studying the microbiome and its role in health and disease [10]. Research on the microbiome

has expanded dramatically in the last decade, with increasing interest in microbial community interactions with cancer.

As an emerging field, challenges must be overcome at all facets of research to ensure robust and rigorous science, and these challenges are only exacerbated by the diversity of the human microbiome. Multiple and concerted efforts have been made to identify and provide solutions to these challenges. The MicroBiome Quality Control project (MBQC) attempted to identify the most critical aspects in microbiome studies to improve reproducibility [14], and the International Human Microbiome Standards consortium (IHMS) attempted to address reproducibility concerns by providing standard workflows for microbiome studies [15]. Several reviews have covered issues and solutions for various levels of microbiology research, including fecal DNA extraction [16], 16S rRNA gene analysis and study design [17], and host-microbe multi-omic analyses [18]. These approaches are eminently worthwhile; though it is important to note, they are continuously evolving as the technology and our understanding of the underlying biology improve. In this review, we address current research and issues in targeting cancer as a disease influenced by the microbiome, which includes the issues of microbial studies addressed above but also specific to correlating microbial analyses with cancer pathology or treatment.

### Historical relationships between the microbiome and cancer

Various microbial populations have been implicated in cancer. In 2002, 17.8% of all cancers were attributed to microbial action [19]. An early causal relationship between a specific bacterial species and human cancer is *Helicobacter pylori* and gastric cancer. *H. pylori* was discovered and later found to be implicated in ulcers by Warren [20]. The development from an *H. pylori* infection to eventual carcinogenesis has been codified in the Correa pathway. *H. pylori* can drive chronic inflammation, which leads to atrophic gastritis and eventual dysplasia. CagA-positive *H. pylori* is especially carcinogenic [21, 22]. More recently, a possible relationship between *H. pylori* in the gut and increased risk of pancreatic cancer has been explored, although it remains controversial [23]. Curiously, *H. pylori* may have a protective effect with respect to esophageal adenocarcinomas [24]. Gastroesophageal reflux disease (GERD) can potentially lead to Barrett's esophagus—that is, a development of scar tissue, cellular dysplasia, and alteration of the cells lining the esophagus from squamous cells to those resembling columnar mucosal cells. These are contributing factors to the development of esophageal adenocarcinoma. There is an inverse correlation between patients with *H. pylori* infections and Barrett's esophagus, and thus with esophageal adenocarcinoma, likely due to the reduction in GERD symptoms as a result of *H. pylori* reducing the local pH in the subregions of the stomach; thereby, the hypothesis goes, reducing the severity of GERD [25]. Thus, a single microbe may have both tumor-suppressing and tumorigenic effects, and deeper research into the host and microbiome relationship is necessary to understand the mechanisms that permit these differing phenotypes.

Transformation-competent viruses have also been shown to cause or be associated with cancer, as was first elucidated through the involvement of Rous sarcoma virus (RSV) in avian sarcoma. RSV is a retrovirus that contains a slightly modified *src* gene that causes the gene product to be unregulated, which modifies intracellular processes

and eventually causes sarcomas in chickens [26]. Human papillomavirus (HPV) has been found to cause cancer by producing the transforming proteins E6 and E7, which prevent Rb from binding E2F and lead to cell cycle dysregulation [27]. Epstein-Barr virus (EBV), a common dsDNA herpesvirus, has been shown to be associated with carcinogenesis, especially Burkitt's lymphomas. EBV infection alone is not sufficient to cause cancer but may lead to carcinogenesis in tandem with genetic and environmental factors [28]. In the case of breast cancer, there was early suspicion that breast cancer in humans may be driven in part by a mammary tumor virus [29]. While this is a known phenomenon in mice, no such virus has been conclusively identified in humans.

### Modern research on the microbiome

Assessing studies of microbial communities and their interactions with cancer can be difficult as there are many methods to look at these interactions, and, occasionally, the approach used in a given study is not entirely clear. Here, we will describe studies by differentiating the relationship between cancer and microbial communities into three categories: primary, secondary, and tertiary interactions. We are proposing this descriptive nomenclature as a means of clarifying exactly what a given study is assessing, as the relationships can occasionally be unclear. We will define the primary interaction as the interaction between a tumor of interest and the microbiota in the local tumor microenvironment (Fig. 1a). Studies done at this resolution are likely looking for direct mechanistic or causal relationships between the microbiota and the tumor, or therapies within the tumor environment, and often require the use of animal models. Recent mouse studies demonstrating that localized bacteria may modulate chemotherapy efficacy are examples of a primary relationship between the microbiota and tumor [30, 31]. Secondary interactions are defined as those between the microbiota involved with the more general tissue or organ environment and the tumor of interest (Fig. 1b), such as the relationship between the gut microbiota from stool and colorectal cancer (CRC). The distinction between primary and secondary interactions is important because, while studies relying on the primary microbiota may elucidate causal relationships, studies of the secondary microbiota might be less capable in this regard due to the relative dilution of cancer-specific interactions in the more generalized microbial population being evaluated. The secondary microbial communities from these sources may contain some signal in the form of traces and residues from the tumor microenvironment and the primary microbiota, but these signals are inherently noisy since they interact with other tissues besides the neoplasm. However, since samples containing the secondary microbiota are much easier to obtain (e.g., stool), this interaction is critical to study in order to identify biomarkers for disease. Tertiary interactions are those where the effect on a tumor or tumor outcome occurs while the tumor is in an entirely different bodily location than that of the microbial community of interest (Fig. 1c). In the vast majority of cases of tertiary interactions, the microbial community is the gut or stool microbiota and the tumors are those outside the digestive tract—for instance, the interactions seen between breast cancer and the stool microbiota or melanoma and the gut microbiota [32–36]. Tertiary interactions often provide strong clinical implications for treatment options but may also afford insight into systemic relationships between the tumor and a physiologically remote microbial community.

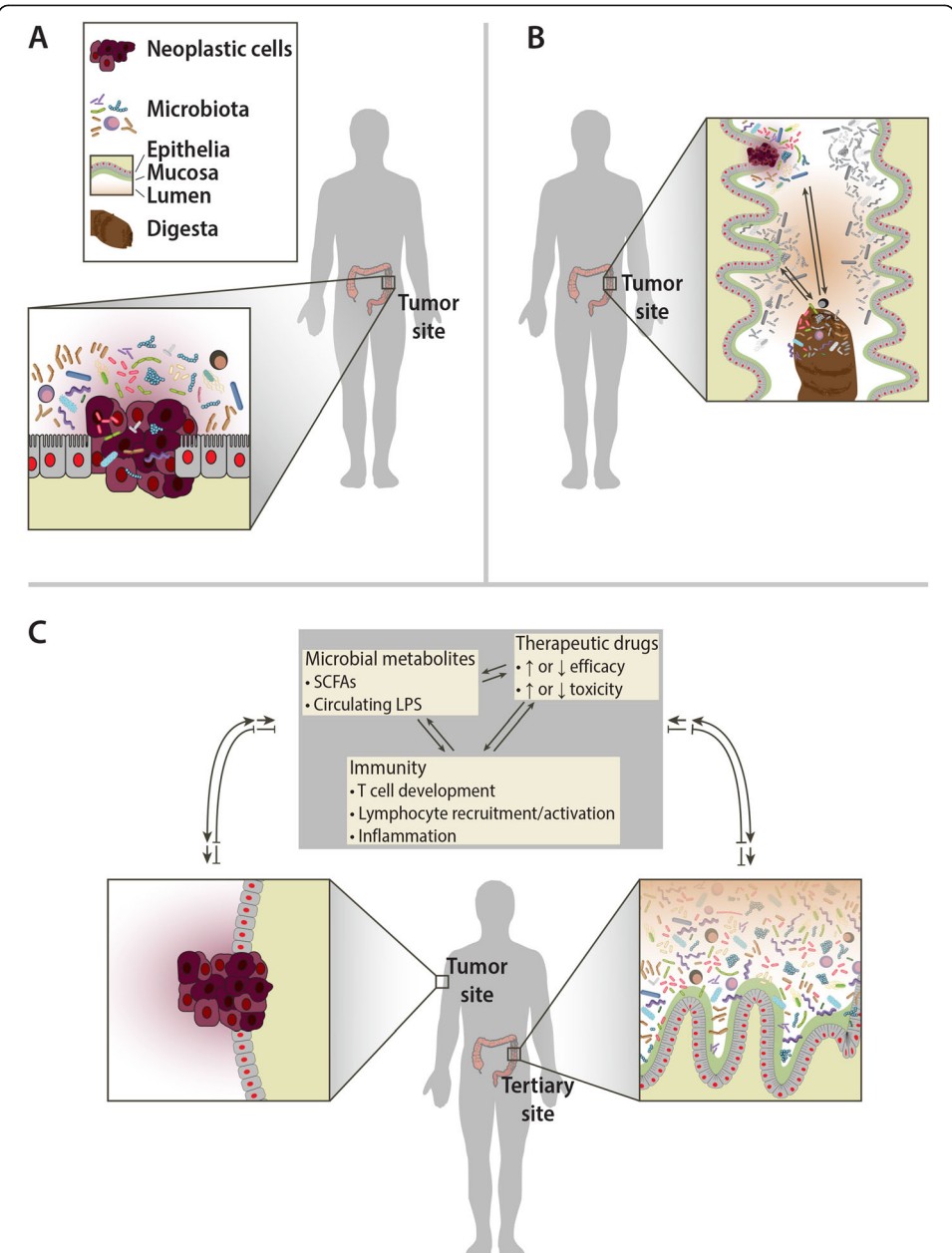

**Fig. 1** Primary, secondary, and tertiary interactions between the tumor and microbial communities. **a** Primary interactions are the interactions within the proximal tumor microenvironment. These interactions are important for understanding the mechanisms of microbiome-cancer relationships, such as tumorigenesis through specific microbes or bacterial proliferation in the tumor microenvironment. **b** Secondary interactions are the interactions between tumors and the microbial community of the tissue or organ system within the same general compartment. These interactions are mostly relevant for discerning potential biomarkers for screening. In this figure, the example is digesta passing by neoplastic tissue in the gut. The digesta may pick up some of the microbes from the tumor, which can be used as a signal of the tumor. Depending on the type and location of the tumor, these interactions may be more or less useful. Generally, an advantage to these interactions is ready access to the material for diagnosis. **c** Tertiary interactions are interactions between a tumor and a remote microbiome community. Tertiary interactions are less direct than secondary or primary interactions; they include therapeutic modulation by modifying chemotherapy drugs and reduce or increase effectiveness or toxicity, or immune modulation that leads to relevant immune cell differentiation or reaction, or metabolites that regulate hormones or host metabolism that can affect cancer phenotypes or outcomes. In spite of the physical distance and separate organ systems these microbial communities occupy relative to the tumor, they can have a profound effect on the tumor phenotype, treatment, and outcomes

## Human tumor microbiota primary interactions

The standard starting place for a study of the microbiota associated with a particular disease state is basic characterization. A first-pass characterization needs to be done to identify the specific taxa that are present in normal and disease states and determine what the potential biomarkers or targets for intervention might be. Thus far, this topic has been studied most commonly through cross-sectional clinical studies, which have identified the microbes or sets of microbes that are differentially present/absent or increased/decreased as a function of the disease once all the potentially confounding patient metadata variables have been accounted for. Most of these types of studies have been restricted to cancers in tissues where there is a resident microbial community. However, one interesting side effect of this line of research is the investigation of tissue sites such as the breast, uterus, prostate, and bladder, among others, that were not previously thought to harbor resident microbial communities [37–45].

Regarding biomarkers, primary tumors are not necessarily a good place to begin a study (i.e., looking at the microbes directly on a tumor to detect if a tumor is present poses some logical challenges). However, there are instances where the primary microbiota can provide useful information. One can discern information about the microbes at the tumor and make predictions about patient outcomes, as is done in parallel research on personalized cancer treatments where a tumor genome is sequenced or specific levels of relevant genes measured as a means of identifying how best to proceed in the clinic. For instance, in some cases, pancreatic cancer can be protected from the immune system by the presence of specific cancer-associated microbial communities and is correlated with patient mortality [46, 47].

Tumors provide a unique hypoxic environment for bacterial growth. In 1955, Malmgren and Flanigan demonstrated in a mouse model that the growth of *Clostridium tetani* is favored in the tumor microenvironment [48]. Tumors can develop hypoxic conditions due to the outgrowth of oxygen supply as a result of poor vascularization by tumor-stimulated angiogenesis [49, 50]. This hypoxic and necrotic environment allows for the selective growth of anaerobic bacteria, an important characteristic of the tumor microbiome [51, 52].

As established by the *H. pylori* model, tumorigenesis may occur due to microbial infection. Other microbes have been suggested to play similar roles. The gram-negative genus *Fusobacterium* is associated with the CRC tumor microenvironment [53, 54]. *Fusobacterium nucleatum*-derived tumorigenesis is thought to arise through an opportunistic infection followed by chronic inflammation and immunosuppression, establishing *F. nucleatum* as an opportunistic cancer driver in the microbiome-tumor primary interaction. It also generates bacterial biofilms that increase fitness for tumor-promoting microbial species and complement the hypoxic tumor microenvironment [55–57]. Enterotoxigenic *Bacteroides fragilis* are another gram-negative anaerobic bacterium associated with CRC and is hypothesized to be a driver of tumorigenesis by promoting mutations in host cell genomes. In the tumor microenvironment, besides proliferation under hypoxic conditions, pathogenic *B. fragilis* may secrete proliferative and proinflammatory signals, thus exacerbating tumorigenesis [58, 59].

The tumor microbiome may play a role in tumor resistance to treatment. Gemcitabine is a cytidine analog used as chemotherapy. In pancreatic ductal adenocarcinoma

mouse models, it was found that tumor resistance to gemcitabine is mediated by intra-tumoral gammaproteobacteria and *Bifidobacterium psuedolongum* [30, 60].

### Microbes as anticancer therapy

A tumor's hypoxic microenvironment's selectivity for anaerobic bacteria has been exploited as a method of combating cancer using auxotrophic bacteria to target the tumor in bacteria-mediated cancer therapy (BCT). Early experiments using *Clostridium* and *Salmonella* alone have not yielded regression of tumors in clinical trials, in spite of success in mouse models. More recently, in various xenograft models, combination therapy using *Clostridium novyi* (*C. novyi-NT*) with chemotherapy, antivascular therapy, or radiation has shown to be more effective than treatment without *C. novyi-NT*, and auxotrophic *Salmonella typhimurium* has shown promising effects. These treatments function by eliciting a local inflammatory response to the bacteria via local innate immune recognition of the bacteria, thereby increasing immune effects on the cancer where the bacteria have localized. Challenges in bacteriolytic therapy revolve mostly around successful bacterial growth in the heterogeneous tumor microenvironment and sufficient stimulation of host immunity to effectively attenuate the tumor [61]. Instead of relying on innate immunity to recognize and react to the localized bacteria, an alternative is to use the bacterial localization to deliver anticancer agents to the tumor environment. Normal cells will be minimally affected, as the bacteria will not localize anywhere but the tumor microenvironment. In in vivo models, this method has shown results in delivering cytotoxic chemotherapy and immune-stimulating drugs to the tumor, although it has yet to prove efficacious for long-term effects. However, this may potentially be compensated for in combination with treatment strategies [51, 62].

The treatments above rely on transferring live bacterial communities, but other treatment methods have been developed using bacterial extracts. Bacterial extracts have been used in cancer treatments since the discovery of Coley's toxins, a concoction of bacterial lysates, in the early twentieth century [61]. More recently, synthetic CpG oligodeoxynucleotides (ODN), small ssDNA segments containing CpG motifs, have been used to stimulate Toll-like receptors (TLRs) on immune cells. They have been classified into three distinct classes: class A stimulates peripheral dendritic cells to secrete type I interferon, class B stimulates B cell maturation, and class C has the effects of both class A and B CpG ODNs [63]. In many models, CpG ODNs are administered intratumorally or locally [64, 65], which can be an issue for administration in some clinical settings. Systemic administration in mice shows promise, however, only when part of combination therapies [66]. At the time of writing, there are multiple ongoing clinical trials for many CpG ODNs, mostly phase 1 or 2 trials for melanomas or lymphomas administered in combination with monoclonal antibody immunotherapies (Table 1) [67, 68].

### Using the microbiome as a biomarker—a secondary interaction

Early detection of cancer increases survival rates [69]. In developing biomarkers, it is imperative to keep in mind that the tools must have robust sensitivity and specificity to reduce false positives and false negatives. Since some microbial samples are minimally/non-invasive to collect in routine checkups, such as the microbiome from the skin,

**Table 1** Ongoing (recruiting or active) clinical trials for CpG ODNs. Results were generated by searching ClinicalTrials.gov with queries "Cancer" and X, where X was "CpG," "SD-101," "IMO-2125," "MGN1703," "SMP-001," "CSI-2," or "GNKG168." Trials were filtered for recruiting, active not recruiting, or enrolling by invitation

| NCT number | Sponsors | Study location | Disease | CpG ODN | Combination therapy | Phase |
|---|---|---|---|---|---|---|
| NCT03831295 | Ronald Levy Bristol-Myers Squibb | Stanford Cancer Institute Palo Alto, Palo Alto, CA, USA | Malignant solid neoplasm | SD-101 | BMS 986178 (anti-OX40) | 1 |
| NCT03410901 | Ronald Levy National Cancer Institute (NCI) | Stanford University, School of Medicine, CA, USA | Lymphoma | SD-101 | BMS 986178, radiation | 1 |
| NCT02927964 | Robert Lowsky Janssen, LP National Institutes of Health (NIH) | Stanford University, School of Medicine, CA, USA | Lymphoma | SD-101 | Ibrutinib (BTK inhibitor), radiation | 1, 2 |
| NCT03007732 | Lawrence Fong Prostate Cancer Foundation Merck Sharp & Dohme Corp. Dynavax Technologies Corporation | University of California San Francisco, San Francisco, CA, USA | Prostatic neoplasm | SD-101 | Pembrolizumab, androgen deprivation therapy, radiation | 2 |
| NCT04050085 | University of California, Davis National Cancer Institute (NCI) Bristol-Myers Squibb Dynavax Technologies Corporation | University of California Davis Comprehensive Cancer Center, Sacramento, CA, USA | Metastatic/refractory pancreatic cancer, stage IV pancreatic cancer | SD-101 | Nivolumab (anti-PD-1), radiation | 1 |
| NCT02521870 | Dynavax Technologies Corporation Merck Sharp & Dohme Corp. | 45 study locations | Metastatic melanoma, head neck cancer | SD-101 | Pembrolizumab (anti-PD-1) | 1, 2 |
| NCT03322384 | University of California, Davis | University of California Davis Comprehensive Cancer Center, Sacramento, CA, USA | Lymphoma, advanced solid tumors | SD-101 | Epacadostat (IDO1 inhibitor), radiation | 1, 2 |
| NCT01042379 | QuantumLeap Healthcare Collaborative | 26 study locations | Breast neoplasms, tumors, cancer, angiosarcoma | SD-101 | Pembrolizumab (anti-PD-1) | 2 |
| NCT03438318 | Checkmate Pharmaceuticals Novella Clinical | 5 study locations | Non-small cell lung cancer | CMP-001 | Atezolizumab (anti-PD-L1), radiation | 1 |
| NCT03507699 | Sheba Medical Center Checkmate Pharmaceuticals Bristol-Myers Squibb | Sheba Medical Center, Ramat Gan, Israel | Malignant colorectal neoplasms, liver metastases | CMP-001 | Nivolumab, ipilimumab, radiation | 1 |
| NCT02554812 | Pfizer | 94 study locations | Advanced cancer | CMP-001 | Avelumab (anti-PD-L1), utomilumab (4-IBB agonist), PF-04518600(OX40 agonist), PD 0360324 (anti-CSF1) | 2 |

**Table 1** Ongoing (recruiting or active) clinical trials for CpG ODNs. Results were generated by searching ClinicalTrials.gov with queries "Cancer" and X, where X was "CpG," "SD-101," "IMO-2125," "MGN1703," "SMP-001," "CSI-2," or "GNKG168." Trials were filtered for recruiting, active not recruiting, or enrolling by invitation (Continued)

| NCT number | Sponsors | Study location | Disease | CpG ODN | Combination therapy | Phase |
|---|---|---|---|---|---|---|
| NCT03618641 | Diwakar Davar<br>Checkmate Pharmaceuticals<br>University of Pittsburgh | UPMC Hillman Cancer Center,<br>Pittsburgh, PA, USA | Melanoma, lymph node cancer | CMP-001 | Nivolumab | 2 |
| NCT03084640 | Checkmate Pharmaceuticals | 4 study locations | Malignant melanoma | CMP-001 | Pembrolizumab | 1 |
| NCT02680184 | Checkmate Pharmaceuticals | 13 study locations | Melanoma | CMP-001 | Pembrolizumab | 1 |
| NCT03865082 | Idera Pharmaceuticals, Inc.<br>Bristol-Myers Squibb | 3 study locations | Solid tumors | IMO-2125 | Nivolumab (anti-PD1),<br>ipilimumab (anti-CTLA-4) | 2 |
| NCT03445533 | Idera Pharmaceuticals, Inc.<br>Bristol-Myers Squibb | 102 study locations | Metastatic melanoma | IMO-2125 | Ipilimumab | 3 |
| NCT02644967 | Idera Pharmaceuticals, Inc. | 10 study locations | Metastatic melanoma | IMO-2125 | Ipilimumab, pembrolizumab<br>(anti-PD-1) | 1, 2 |
| NCT04126876 | A.J.M. van den Eertwegh<br>Idera Pharmaceuticals, Inc. | VU Medical Center, Amsterdam,<br>the Netherlands | Malignant melanoma | IMO-2125 | | 2 |
| NCT02077868 | Mologen AG | 127 study locations | Metastatic CRC | MGN1703 | Usual maintenance | 3 |
| NCT02668770 | M.D. Anderson Cancer Center<br>Mologen AG | University of Texas MD Anderson<br>Cancer Center, Houston, TX, USA | Advanced cancers, melanoma | MGN1703 | Ipilimumab | 1 |
| NCT02452697 | David Rizzieri, MD<br>Agilent Technologies, Inc. | Duke University Health System,<br>Durham, NC, USA | Myeloid malignancies,<br>lymphoid malignancies | DUK-CPG-001 | NK-enriched donor lymphocyte<br>infusion (DLI) | 2 |

mouth, rectal swabs, and urine, they can provide facile sources of investigation to assess as cancer biomarkers.

Microbes involved in the primary tumor microenvironment are capable of becoming biomarkers found in secondary samples. In practice, this depends on a few factors. For one, there is the location of the tumor. Tumors that are in a region of the body exposed to the environment or involved with circulated or excreted substances, such as cancers of the blood or digestive system, have microbial biomarkers that are easier to obtain since they may be detected in the blood, feces, etc. Another important factor is the auxotrophy of the microbes—they may be dependent on the tumor microbiome primary environment and thus not found at detectable levels elsewhere in the body.

Current clinical screening methods for CRC have shortcomings. Some tests that are sensitive are invasive or expensive, and other tests that are non-invasive or inexpensive have low specificity or sensitivity [70]. There remains a need to find non-invasive, effective, and cost-effective screening tools for CRC, and characterization of the microbiome may improve clinical screening and risk assessment. A pilot study has shown that progressive stages of colorectal cancer, labeled as having no malignancy, adenoma, and carcinoma, can be identified based on the relative abundance of microbes in fecal samples. Combining this method with a fecal occult blood test (FOBT) decreased the chance of false positives [71]. Several specific bacteria have been associated with CRC that may be promising targets for developing biomarkers. As mentioned above, *Fusobacteria* is present in primary colorectal tumor interactions; however, using species-level *Fusobacteria* taxa as biomarkers has proved challenging. *F. nucleatum* may be a potential driver of tumorigenesis in the primary interaction, but it is only present in a minority of CRC patients and is also present in the stool of non-CRC individuals [56]. *F. nucleatum* is also present in the oral microbiome, and unusual levels of this bacterium in the mouth may be a biomarker of CRC, an example of a tertiary interaction as a biomarker for cancer [72].

Pancreatic cancer is one of the most fatal cancers with a 5-year mortality between 91 and 98%. A large contributor to the high mortality is the lack of efficient screening methods [73]. Pancreatic cancer, like many cancers, is largely driven via inflammation. Similar to gastric cancer, *H. pylori* infection increases the risk of pancreatic cancer. Periodontal disease also leads to an increased risk of pancreatic cancer. Studies thus far have found multiple associations between various oral microbiome perturbations and pancreatic cancer [74, 75].

In addition to those described here, there is a similar clinical need for biomarkers in other cancers. There have been a multitude of studies looking at the microbial communities in secondary compartments for relevant genomic markers, including the head and neck, lung, bladder, prostate, and others [43, 45, 76–78]. The stool microbiome has also been used as a tertiary biomarker for distant tumors, including hepatocellular carcinoma, lung cancer, and prostate cancer [78–82].

## Modulation of therapeutic drug efficacy by the microbiome—a tertiary interaction

Chemotherapy, a common and effective means of treating many cancers, involves using drugs that preferably destroy transformed cells with minimal damage to normal cells, although side effects may be common. Bacteria in the microbiome have been shown to

modulate chemotherapy in various capacities. The most obvious effect is the microbial effect on orally administered drug metabolism. Microbial species lining the intestinal mucosa can perform various metabolic reactions, such as reduction, hydrolysis, ring opening, and functional group removal. These alterations can activate or deactivate drugs, or affect drug toxicity [83]. Alternatively, these microbes can affect intestinal cell gene expression to modulate host cell metabolism of the drugs, which has implications for drug absorption and efficacy over time [84, 85] The direct clinical relevance of gut bacterial modulation of drugs is apparent in the case of irinotecan (CPT-11) treatment, where the presence of bacteria in the gut expressing β-glucuronidase increases the CPT-11 side effect of severe diarrhea [85, 86].

The microbiome has modulatory effects in other therapies. For allogeneic hematopoietic cell transplantation (allo-HCT), a treatment for hematologic cancers, the gut microbiome may be predictive for both risk for treatment and chances of relapse post-treatment [87, 88]. In the case of inflammation-based immunotherapy, such as intratumoral administration of CpG ODN described above, an intact gut microbiome was found to significantly increase treatment efficacy in mice [89]. TLR response-based therapeutics are connected to innate immunity, which may generally be affected by the host microbial communities, although the specific mechanisms remain unknown. Other therapeutic agents that do not rely on innate immunity have also been studied. Cisplatin and oxaliplatin, platinum-based chemotherapy that forms intrastrand adducts and interstrand crosslinks to prevent polymerase activity, were found to have reduced efficacy when combined with common antibiotics in mice [89]. Cyclophosphamide (CTX) is a chemotherapy drug that alkylates and crosslinks DNA and similarly prevents polymerase activity. CTX is known to be immunogenic, meaning through cell lysis via CTX, tumor cells release immune-stimulating signals that allow dendritic cells (DC) to target the tumor, thus increasing CTX efficacy [90]. In mice, Viaud et al. found that CTX also leads to increased numbers of pathogenic $T_H17$ and increased $CD4^+$ T cell differentiation to $T_H1/T_H17$ by modulating the gut microbiota. However, the use of antibiotics along with CTX was found to decrease CTX efficacy [91].

The efficacy of the checkpoint inhibitors PD-1, PD-L1, and CTLA-4 is modulated by the gut microbiota. In multiple melanoma GF mouse models, the presence of specific microbial taxa along with checkpoint inhibitors leads to an increased presence of relevant immune-activating $CD4^+$ and $CD8^+$ T cells in the tumor microenvironment and DC IL-12 secretion [92–94]. These studies can have significant clinical relevance, as the gut microbial composition is a likely determinant of patient response or non-response to anti-PD-1 and anti-PD-L1 therapy [32, 95]. While microbial-mediated immune stimulation can have a positive effect on the tertiary tumor, it can also have a negative effect by driving immune checkpoint inhibitor (ICI)-associated colitis. Colitis has been shown in mouse models to drive primary interactions that can allow genotoxic bacteria to ingraft and eventually lead to dysplasia [96]. This issue is manageable, however, given that patients who develop or did not develop colitis after anti-CTLA-4 therapy have distinct microbial communities [35], $IL-10^{-/-}$ mouse models show *Bifidobacteria* may prevent colitis after anti-CTLA-4 therapy [97], and FMT may prove effective [98]. Thus, analysis of a patient's gut microbiota may provide insight into the checkpoint inhibitor treatment efficacy for a remote tumor and risk assessment for ICI-associated colitis.

In general, dysbiosis or altered states of microbial communities may result from a patient's genotype, antibiotic exposures, or chemotherapy, and altering or restoring microbiota may lead to increased immune responses via innate immune signaling (TLR) or specific immune cell activation or differentiation, e.g., $T_H1/T_H17$, $CD8^+$ T cells, or DC [51]. There is emerging interest and effort being put forth to model the interactions between the immune system, therapeutic interactions, and cancer outcomes with the microbial communities in the host as the key variable [91, 99–103].

## Future directions

Culture-independent approaches have become the forefront of research on the cancer microbiome. As sequencing has become much less expensive and the 16S rRNA gene databases have expanded, 16S rRNA gene sequencing has become an efficient and high-throughput method for identifying bacteria present in a microbial community [104, 105]. Issues with the resolution of 16S rRNA gene sequencing at the species and strain levels remain a challenge [105, 106]. With the development of second-generation and third-generation sequencing, one method to overcome the shortfalls of 16S rRNA gene sequencing is to simply sequence the entire metagenome. Whole-genome shotgun (WGS) sequencing, especially WGS techniques that provide long reads, has the ability to overcome the 16S rRNA gene sequencing limitations [104, 107, 108]. By analyzing the entire genome of all the microbial populations present, greater microbial diversity can be represented with greater accuracy than 16S rRNA gene sequencing. Viral and fungal species can be represented as well [109]. While advances in these technologies are making WGS sequencing become more affordable, the cost to accurately sequence and analyze all the bacteria in a microbial population with sufficient coverage remains expensive, and the data analysis is laborious and intensive [107, 109] Progress is being made on these fronts as well, with researchers leveraging advances in the algorithms used to identify species- and strain-level details to allow shallow WGS to be used to gain the advantage of the resolution of WGS with the economic benefit of 16S rRNA gene sequencing [110].

Generally speaking, microbiome studies each come with their own caveats and challenges. The Human Microbiome Project was the first large-scale effort to catalog the microbial communities present across multiple body sites [10, 111–114]. These studies, along with other, similar efforts [115], have acted as the cornerstone of human microbiome work as they allow researchers to compare and contrast their own work on the cancer microbiome with the research generated by others. However, while conceptually simple, in practice, these comparisons are plagued with confounding variables that hamper clear comparisons (Fig. 2 and Table 2).

The work of Sinha et al. has evaluated many experimental factors that influence the evaluation of microbial communities [14]. Their work is part of a larger effort to identify and iterate upon "best practices" in the microbiome research field. For example, the specific microbial taxa detected in samples vary as a function of which region of the 16S rRNA gene is targeted for PCR amplification and sequencing (e.g., V1–V3 vs. V3–V5) [10, 14, 124]. Subsequent research by other groups has also uncovered many additional variables that can influence the results of 16S rRNA gene sequencing studies, including the use of sample preservatives [116–122], sample storage [117–119, 129, 132–142], sample lysis protocol [143–145, 149, 150], and DNA purification technique [116–156]. The

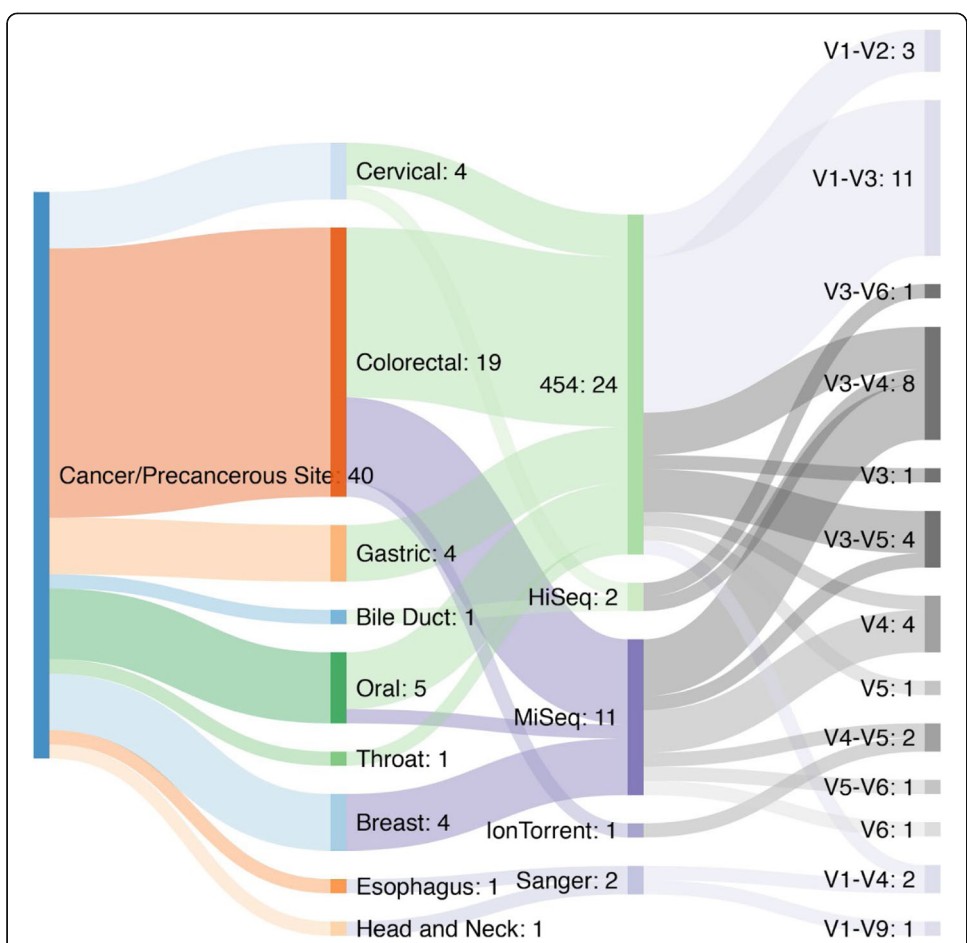

**Fig. 2** Studies of primary interactions between tumors/precancerous lesions and bacteria are diverse with respect to the sequencing platforms used and the 16S rRNA gene variable regions targeted for PCR and sequencing. This Sankey plot highlights that among 40 different studies of the primary microbial communities found at the site of a lesion, there are 5 different sequencing technologies that have been used and 13 different sets of variable regions targeted

**Table 2** Several variables related to the collection and evaluation of microbial communities by amplicon sequencing have been performed. This table presents a subset of relevant variables and references to the studies in which they have been evaluated

| Variable evaluated | Study references |
| --- | --- |
| Use of preservative agents (cryoprotectants, RNAlater, etc.) | [116–123] |
| Sequence database | [124–128] |
| Sequencing platform | [116, 125, 129–131] |
| Sample storage | [117–119, 127, 129, 132–142] |
| DNA extraction method (comparison of kits) | [116, 125, 127, 129, 130, 132, 140, 143–148] |
| Lysis method (chemical, enzymatic, mechanical) | [143–145, 149–151] |
| Sample collection (method, sampling location, sampler) | [118, 126, 129, 131, 134, 150, 152–156] |
| 16S rRNA gene variable region sequenced | [10, 14, 124] |

choice of PCR primers, even for the same variable region; the DNA polymerase used; the PCR cycling conditions; and the analysis platform can influence the outcome of 16S rRNA gene sequencing studies [129, 157–159]. These variables, combined with the additional factors such as bioinformatic pipelines used for analysis, data quality control tuning, software parameters, and reference database choices, raise many questions related to the value of comparing the *final* results of these studies to one another, when none of these factors are controlled for. One would hope that the results from one bioinformatic analysis platform to another would be robust and adequately reflect the biological reality of the experimental system; however, this has been demonstrated by several groups to not necessarily be the case [159–165]. The development of new tools and the constant updating of existing tools are proceeding rapidly, with a healthy amount of back and forth discussion between different research groups (e.g., see Edgar [166] and the associated pre-publication comment by the QIIME development team).

The body of microbiome studies on cancer as a whole manifests as one might expect a decentralized, multidisciplinary research endeavor would: there is a profound lack of consistency regarding the analyses. Figure 2 is a Sankey plot that summarizes the tumor types, sequencing technologies, and the 16S rRNA gene variable region targeted across 40 different studies of the primary interactions between tumors and their associated bacteria [39, 40, 54, 167–203]. While not shown, these studies also vary with respect to the data analysis tools, pipelines, and databases used. Of these, more than half (26/40) did not provide sufficient information in the methods section or supplementary materials to ascertain what the bioinformatic approaches used were in sufficient detail to be able to replicate their analyses. Of the others that indicated the tools and approaches used, at least seven different tools or combinations of tools were used. Furthermore, the specifics of the parameters applied were not sufficiently reported. The studies included in Fig. 2 are also quite varied with respect to the availability of the raw sequence data. From an assessment of the manuscripts and their accompanying supplementary materials, as well as a cold search of the NCBI Sequence Read Archive (SRA) for the studies themselves, 27 of the 40 have deposited their data in the SRA, in some form or other, 1 of the 40 deposited their data in the European Nucleotide Archive (ENA) and 12 of the 40 include a statement that data are available upon request or do not mention data availability at all. One study has the data deposited in a public database, but not the associated metadata that would allow other researchers to replicate their findings, specifically due to the manner in which the IRB protocol for this study was written. In order to improve the ability of researchers to replicate published findings, journals have begun to require that data underlying figures as well as the code used to generate statistical models and visualizations be made available as supplementary information, though even with published code and tools, there is a broad spectrum of usability [204]. Code generated and used in publications is often made available on GitHub or SourceForge. The journal eLife has made code persistently available by forking the authors' published version of code directly to the eLife GitHub account. Challenges remain regarding software and database availability, however. There are efforts in the field to provide bioinformatic virtual environments in the form of virtual machine (VM) images or dockerized containers that would allow interested researchers to be able to easily repeat the analyses shown in a publication, as these environments contain the exact software and database versions used at the time—QIIME 2, a popular

metagenomic analysis suite, is provided in a preconfigured VM [205]. Even in these best-case scenarios, this distribution model is suboptimal when the tools or databases used are proprietary or closed source and not distributable. Despite these challenges, steps are being taken in the field to develop guidelines for computational tool development and publication [206].

The field of microbiome research is still new, and it is understandable that there would be diversity in data analysis approaches and reporting. Unfortunately, as the field of biomedical research struggles with a reproducibility crisis, it is important that these deviations are addressed [207]. To deal with these challenges, several groups have provided guidelines that researchers can utilize to improve the utility and robustness of their findings, including MIMARKS and MIMAG for reporting sequencing data from microbes [208, 209], STROBE and STROME-ID for molecular data tied to epidemiological projects [210, 211], and REMARK for data used as tumor markers [212, 213]. Marques and colleagues, working in the field of hypertension, have published a set of general guidelines and checklists that researchers and publishers can use when designing, executing, and reporting microbiome-related studies of disease in either animal models or humans that could easily be adapted to cancer research [214]. Other groups, including the International Human Microbiome Standards Project and the Microbiome Quality Control project, have proposed research and reporting guidelines for experimental design when assessing the microbiome in humans and model systems [14, 215, 216].

The many variables related to research on cancer-associated microbiota have led other research groups to attempt to integrate findings across studies in meta-analyses, typically focused on CRC and other gut-related diseases [198, 217–219], while other reviews on the topic have attempted to integrate the non-gut-related microbial changes [220]. In the case of the former, a recent meta-analysis that was performed on the colorectal cancer microbiome [219] found that when integrating data from multiple projects, many of the significant factors that were reported in individual studies were discovered to *not* be generalizable (i.e., the relationship between the increased abundance of *Fusobacterium* ssp. in CRC patients). In the case of the latter, the findings documented in these reviews are intriguing—however, as the comparisons are massively confounded by the indicated variables mentioned above, they are challenging to rely upon as a basis for future work.

As the field generates more and more studies of the microbial communities relevant to cancer biology, it is common (and quite reasonable) for researchers and reviewers to want to contextualize findings in light of what has already been done on the topic. At the moment, this is done ad hoc either by comparing the final results of other projects to one's own work or by more quantitatively integrating the work of other researchers into the analyses using either raw sequencing data or finalized OTU tables, when available. Standardizing methods and protocols across the field of cancer, microbiome research will prove useful in the short run, but since tools will change and evolve, robust controls, logical study design, and sparing no effort to document and report on research materials will be vital to the field.

## Conclusions

The promise of the microbiome in cancer research is tremendous. The field is rapidly expanding in scope along with our understanding of how microbial communities that

live in and on us shape our behavior and influence our health. These breakthroughs in microbiome research have only been made possible by the advances in genomics—these advances have allowed us to collect vast amounts of metagenomic and microbial marker gene sequences. Studies have now characterized the essential role of the microbiota in tumor formation and fitness. We have highlighted several areas that show prospective benefits for cancer patients in the clinic. Characterization of the tumor microenvironment led to the use of microbes that are selective for the said microenvironment in therapies. Current trials are exploring the use of bacterial extracts in combination with checkpoint inhibitor immunotherapies. A deeper understanding of the microbiome in the context of secondary interactions brought about research into useful diagnosis and detection of cancer early in the process. We have also shown that continued, efficient research in the emerging field of the microbiome in the context of cancer is predicated on identifying best practices surrounding the design, execution, bioinformatic analysis, interpretation, and reporting of studies. Due to the lack of applied common experimental and analytical approaches for research methods and subsequent extreme variability in study design, outlining overarching, interpretable themes between multiple studies even within the same cancer type, much less across cancer types, is challenging to say the least. We propose an update to the manner in which the interactions between cancer and microbial communities are described to rectify these inconsistencies and allow for larger-scale interpretations from multiple studies. Our hope is that the temporary shortcomings in microbiome research will be recognized and accounted for in order to provide the research community, clinicians, and patients with the best possible information and outcomes possible.

## Supplementary information

---

**Additional file 1.** Review history.

---

**Peer review information**
Kevin Pang was the primary editor of this article and managed its editorial and peer review in collaboration with the rest of the editorial team.

**Review history**
The review history is available as Additional file 1.

**Authors' contributions**
MBB and SSM co-wrote the first draft of the manuscript. AMD substantially revised the manuscript and coordinated with SSM to generate the tables. All authors contributed to the writing, revision, and preparation of the review. The author(s) read and approved the final manuscript.

**Funding**
None applicable.

**Ethics approval and consent to participate**
Not applicable

**Competing interests**
The authors affirm that they have no competing interests.

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

## 

