## [**Additional file 1.** Review history. · Genome Biology]

Review history

First round

Reviewer 1

Is the topic of the article timely and of interest to a wide range of readers?

Yes

Is the article well written and presented in a logical way?

Yes

Do the authors cover the relevant literature in an accurate and balanced way?

Yes

Do the authors provide a useful synthesis of the topic?

Yes

Do the authors provide insightful discussion of the future directions for the field?

Yes. Yes but while they have many correct critiques about practices in the field, they ignore all the work that has been done to try to improve how this is done, such as guidelines, a published large multicenter quality-control project, etc.

Do any tables, figures or boxes that are present enhance the manuscript?

Yes

Would the manuscript benefit from any additional tables, figures or boxes?

Yes

Manzoor and Burns have written a review article on some of the literature in cancer and microbiota. There are already numerous reviews on microbiome & cancer, and the specific contributions that distinguish this manuscript are a classification of interaction types between microbial species and tumors and a call to action about reproducibility and transparency in microbiome research. The review of the cited literature itself is done accurately, and the prose is well written.

Specific comments:

There should be a sentence somewhere near the beginning that refers to other review articles about microbiome and cancer and highlights for the reader what differentiates this piece from the others that preceded it. The methodological issues that the manuscript discusses should be framed in the context of the review articles that have been previously published on the topic in this journal (and others).

-Greathouse Genome Biology 2019 "DNA extraction for human microbiome studies: the issue of standardization"

-Mallick Genome Biology 2017 "Experimental design and quantitative analysis of microbial community multiomics."

-Debelius Genome Biology 2016 "Tiny microbes, enormous impacts: what matters in gut microbiome studies?"

Throughout, the authors should specify whether the reported result in each study was observed in animal vs. human studies.

The analysis of publication trends in Figure 1 is superficial. Did the authors include variants of terms, such as for microbiome: metagenomic, microbiota, commensal, etc.? And for cancer: neoplastic, carcinoma, lymphoma, chemotherapy, adenocarcinoma, sarcoma, leukemia, etc.? Were review articles and commentaries excluded? The field of literature meta-analysis has rigorous methods that are used for systematic reviews; the relevant techniques for the literature search should be applied if one wants to publish this sort of an analysis of publication trends. However, this figure adds little to the review and an alternative could be to omit it. The point that there is a lot of interest and many papers on the topic can be made without this figure.

The classification of primary, secondary, and tertiary interactions is interesting and useful. If this is one the authors have developed, that should be explained along the lines of "we propose..." If other authors have previously used this nomenclature, that should be cited.

Figure 2 should label the elements or provide a legend to define, for example, that red cells are cancer, that the white space is the gut lumen, that the grey cells with red nuclei are normal gut epithelial cells. In the "secondary" panel would it make more sense for the arrows to point to a non-neoplastic segment of the epithelium, to emphasize the interactions between the bacteria and those cells? Or at least a second pair of arrows going between the bacteria and the normal epithelium? And in the 'tertiary' panel why are the bacteria so close to the tumor if tertiary interactions are defined as the tumor being "in an entirely different bodily location than that of the microbial community of interest"?

Page 10 line 15 cites the fact that many papers have been published about how sample processing affects sequencing results and with a single citation to 40 articles (Ref# 107-147). Simply dumping this list of articles into the bibliography is hardly more useful to the reader than a pubmed search. The authors should summarize the key studies and synthesize the main take-away points, perhaps with a table or just the salient takeaways with a few specific examples in the text. Also, many of these issues were addressed in the multicenter Microbiome Quality Control Project (Sinha Nature Biotechnology 2017) which should be cited, and in the previously published review articles on this topic in Genome Biology that I listed above.

On Page 10 the authors claim that the lack of consistency in analysis methods results from the fact that researchers "tend to be cancer specialists" with limited bioinformatic experience. This is an opinion and not supported by any data. The authors are correct in calling attention to the chaos of analytical approaches and the dearth of methodological details in the literature. However, rather than pointing fingers at the the academic training paths of people engaged in the research, the authors could instead do 3 things:

(a) frame this problem as an example of the broader reproducibility crisis in biomedical research and cite a few of the prominent editorials, journal & funding-agency policies, and code repositories that have been established to address it.

(b) refer readers to the collaborative guidelines that the microbiome & cancer epidemiology fields have promulgated to try to address these problems, such as MIMARKS (Yilmaz Nature Biotech 2011) MIGS (Field Nature Biotech 2008), STROBE (von Elm PLOS Medicine 2007), STROME-ID (Field Lancet Infectious Disease 2014) and REMARK (Altman PLOS Medicine 2012).

(c) if, after familiarizing themselves with these bodies of work, the authors note that despite all of these efforts, the microbiome & cancer field still continues to be plagued by these problems (as do many other fields in biomedicine), they might then offer some suggestions to investigators, funding agencies, journals, and institutions about how to reduce the barriers to & incentivize guideline adherence, how to improve training for investigators, or offer suggestions for future updates to the guidelines or public repositories to make them better & easier to use.

There are some additional papers on cancer therapy & microbiome that should be cited: The work of Gajewsky should be cited along with the work of the Wargo and Zitvogel groups on checkpoint blockade (Sivan Science 2015 and Matson Science 2018). There is a body of literature on bone marrow transplantation as a treatment for cancer (e.g. Taur Blood 2014 and Peled J Clinical Oncology 2017), work by the Jobin group on colorectal cancer (e.g. Arthur Science 2012), and work on the colitis side effects of cancer checkpoint blockade therapy (Wang & Jenq Nature Med 2018, Dubin Nature Communications 2016, and Wang & Davis PNAS 2018)

Page 2: I believe that the discovery of EBV & Burkitt lymphoma preceded the work on H. pylori & gastric cancer. Please confirm the history of research on each well-known etiologic relationship between a microbe & cancer in order to claim that "Perhaps the earliest causal relationship..."

"Obviously" (p 4 line 29) this may not be so obvious, given that microbiota can have systemic effects on e.g. inflammation, which, in-turn, can have effects on a remote cancer, as a "tertiary interaction".

Reference #62 is a set of search terms against the clinicaltrials.gov database, which will be dynamic change over time. Please replace this reference with some sort of a static link, or a table that lists relevant ongoing trials.

The discussion of pros & cons of different screening methodologies for colorectal cancer at the top of page 7 does not contribute much. The point that microbiome profiling might improve colon-cancer screening or risk-assessment can be made without wading into this controversial clinical area.

Minor points:

P5 line 20 " It also facilitates..." can be read as if the bacterium is generating anaerobic conditions. But do you mean the tumor generates anaerobic conditions?

A few colloquial phrases could be made more precise/scientific:

"our bodies" (p 1, line 33)

" these show up in the literature as" (p 4, line 23)

"into your own analyses" (p11, line 47)

" that also look at" (legend to Fig 1)

"chemo" in Figure 2

Typo p5 line 45 " tumore's"

P1 line 57 "suggest" not suggests

Reviewer 2

Is the topic of the article timely and of interest to a wide range of readers?

Yes

Is the article well written and presented in a logical way?

Yes

Do the authors cover the relevant literature in an accurate and balanced way?

Yes

Do the authors provide a useful synthesis of the topic?

Yes

Do the authors provide insightful discussion of the future directions for the field?

No

Do any tables, figures or boxes that are present enhance the manuscript?

Yes

Would the manuscript benefit from any additional tables, figures or boxes?

No. The figures provided are sufficient.

Topic: The authors describe recent developments in the cancer ~ microbiome field. I agree with the authors that this topic is a worthwhile one which should be of interest to a relatively large audience.

Minor Points:

1. It strikes me as odd to start the Introduction with a discussion of the semantic differences of describing the microbiota and the microbiome as physical organisms versus genomic content. I would strongly suggest that the authors reconsider how they would like to best use the unexhausted attention span which the readers will bring to the first paragraph of the manuscript.

2. There are some odd spacing errors in this text. Please check for " . ", for example on line 13.

Major Points:

3. The strength of this paper is its discussion of the biological basis of interaction for cancer and the microbiome. I found the text written by the authors to be engaging, detailed, and informative. However, the topic which I found to be lacking was the discussion of the "genome biology" of this interaction, and the findings of the field generated with genomic methods. There is a good discussion of the underlying methods for 16S, and a bit of WGS, but very little review of the results of these studies and whether there are any common threads which add to our understanding of this biological system. I would prompt the authors to revise this manuscript in such a way that it will answer the question: "Have genomic methods of characterizing the microbiome substantively added to our understanding of the interaction between microbes and the development or treatment of cancer?" The authors clearly have a deep understanding of this field, and I think they would be well positioned to answer such a question.

Responses to Reviewer Comments

Editor

...

As you will see from the reports, both referees are broadly favorable and find the review of potential interest, but they raise important issues that we must ask you to address, in the form of a revised manuscript, before we reach a final decision on publication. We do strongly agree with reviewer 2 about incorporating more about how genomics will help with microbiome research. We don't necessarily think it needs to be more methods/bias focused, but perhaps relating the topics back to genomic perspective could be useful.

...

In line with increasing the commentary on genomic/tying things back to genome sciences, we have extensively revised the text and incorporated additional text and references relating the biomedical findings to the technical challenges faced when performing metagenomic analyses.

Reviewer #1

Manzoor and Burns have written a review article on some of the literature in cancer and microbiota. There are already numerous reviews on microbiome & cancer, and the specific contributions that distinguish this manuscript are a classification of interaction types between microbial species and tumors and a call to action about reproducibility and transparency in microbiome research. The review of the cited literature itself is done accurately, and the prose is well written.

Specific comments:

1. There should be a sentence somewhere near the beginning that refers to other review articles about microbiome and cancer and highlights for the reader what differentiates this piece from the others that preceded it. The methodological issues that the manuscript discusses should be framed in the context of the review articles that have been previously published on the topic in this journal (and others).
 - Greathouse Genome Biology 2019 " DNA extraction for human microbiome studies: the issue of standardization"
 - Mallick Genome Biology 2017 "Experimental design and quantitative analysis of microbial community multiomics."
 - Debelius Genome Biology 2016 " Tiny microbes, enormous impacts: what matters in gut microbiome studies?"

These are excellent suggestions for inclusion and we agree that framing our work in this context will improve this review. We have added these citations as well as added the suggested framing text in the introduction.

2. Throughout, the authors should specify whether the reported result in each study was observed in animal vs. human studies.

This is a clarifying detail that we overlooked - we have made an effort to clarify whether the studies of interest were in humans or a model system where appropriate.

3. The analysis of publication trends in Figure 1 is superficial. Did the authors include variants of terms, such as for microbiome: metagenomic, microbiota, commensal, etc.? And for cancer: neoplastic, carcinoma, lymphoma, chemotherapy, adenocarcinoma, sarcoma, leukemia, etc.? Were review articles and commentaries excluded? The field of literature meta-analysis has rigorous methods that are used for systematic reviews; the relevant techniques for the literature search should be applied if one wants to publish this sort of an analysis of publication trends. However, this figure adds little to the review and an

alternative could be to omit it. The point that there is a lot of interest and many papers on the topic can be made without this figure.

We have two points to make in this regard: first yes, all the variants that you've cited were accounted for (it turns out that the search engine underlying PubMed does an excellent job tagging studies with relevant key terms), and yes, we were looking only at primary literature on the topic. We agree that the text could have been far more substantial in describing exactly how this was generated. In this regard, we were simply attempting some small level of quantitation rather than simply asserting that there have been lots of studies of cancer and the microbiome in recent years. Second, we agree that this figure could simply be summarized/replaced with some short descriptive text - we have done so in the main text of the article.

4. The classification of primary, secondary, and tertiary interactions is interesting and useful. If this is one the authors have developed, that should be explained along the lines of "we propose..." If other authors have previously used this nomenclature, that should be cited.

This is a descriptive approach that we came up with - thank you for the suggestion to make this more explicit. We have made sure to highlight that this is something we are independently proposing.

5. Figure 2 should label the elements or provide a legend to define, for example, that red cells are cancer, that the white space is the gut lumen, that the grey cells with red nuclei are normal gut epithelial cells. In the "secondary" panel would it make more sense for the arrows to point to a non-neoplastic segment of the epithelium, to emphasize the interactions between the bacteria and those cells? Or at least a second pair of arrows going between the bacteria and the normal epithelium? And in the 'tertiary' panel why are the bacteria so close to the tumor if tertiary interactions are defined as the tumor being "in an entirely different bodily location than that of the microbial community of interest"?

These are excellent observations and suggestions for improvement and clarification. We have likely gone over this figure and become somewhat blinded to elements that a new observer might find unclear. We have added a more substantive legend and updated the images and interactions describing secondary and tertiary interactions.

6. Page 10 line 15 cites the fact that many papers have been published about how sample processing affects sequencing results and with a single citation to 40 articles (Ref# 107-147). Simply dumping this list of articles into the bibliography is hardly more useful to the reader than a PubMed search. The authors should summarize the key studies and synthesize the main take-away points, perhaps with a table or just the salient takeaways with a few specific examples in the text. Also, many of these issues were addressed in the multicenter Microbiome Quality Control Project (Sinha Nature Biotechnology 2017) which should be cited, and in the previously published review articles on this topic in Genome Biology that I listed above.

This is a good way to make sure that other researchers are able to track down specific papers that cover the variables they care about. We have updated the manuscript to include both text descriptions for subsets of references (including those that you provided) as well as a table (Table 2) that breaks down the specific factor of interest along with the references that have tested them.

7. On Page 10 the authors claim that the lack of consistency in analysis methods results from the fact that researchers "tend to be cancer specialists" with limited bioinformatic experience. This is an opinion and not supported by any data. The authors are correct in calling attention to the chaos of analytical approaches and the dearth of methodological details in the literature. However, rather than pointing fingers at the the academic training paths of people engaged in the research, the authors could instead do 3 things:
 - a. frame this problem as an example of the broader reproducibility crisis in biomedical research and cite a few of the prominent editorials, journal & funding-agency policies, and code repositories that have been established to address it.
 - b. refer readers to the collaborative guidelines that the microbiome & cancer epidemiology fields have promulgated to try to address these problems, such as MIMARKS (Yilmaz Nature Biotech 2011) MIGS (Field Nature Biotech 2008), STROBE (von Elm PLOS Medicine 2007), STROME-ID (Field Lancet Infectious Disease 2014) and REMARK (Altman PLOS Medicine 2012).
 - c. if, after familiarizing themselves with these bodies of work, the authors note that despite all of these efforts, the microbiome & cancer field still continues to be plagued by these problems (as do many other fields in biomedicine), they might then offer some suggestions to investigators, funding agencies, journals, and institutions about how to reduce the barriers to & incentivize guideline adherence, how to improve training for investigators, or offer suggestions for future updates to the guidelines or public repositories to make them better & easier to use.

Again, these are very helpful suggestions that we have made sure to address in the revised manuscript. We have highlighted that this is not something unique to just one field, removed any non-quantitative observations about specialties, highlighted the excellent work that has and is being done to promote standardization in design, execution, and reporting of data and code, and discussed some of the efforts being made outside of research labs to streamline standardization.

8. There are some additional papers on cancer therapy & microbiome that should be cited: The work of Gajewsky should be cited along with the work of the Wargo and Zitvogel groups on checkpoint blockade (Sivan Science 2015 and Matson Science 2018). There is a body of literature on bone marrow transplantation as a treatment for cancer (e.g. Taur Blood 2014 and Peled J Clinical Oncology 2017), work by the Jobin group on colorectal cancer (e.g. Arthur Science 2012), and work on the colitis side effects of cancer checkpoint blockade therapy (Wang & Jenq Nature Med 2018, Dubin Nature Communications 2016, and Wang & Davis PNAS 2018)

We have incorporated the additional relevant citations mentioned above in the revised manuscript in the section covering checkpoint inhibitors. We are grateful for the suggestions as it provides credit where it belongs and has improved the manuscript.

9. Page 2: I believe that the discovery of EBV & Burkitt lymphoma preceded the work on H. pylori & gastric cancer. Please confirm the history of research on each well-known etiologic relationship between a microbe & cancer in order to claim that "Perhaps the earliest causal relationship..."

This is an excellent suggestion for clarification. We clarify by removing the superlative claim and specify by mentioning "a specific bacterial species and human cancer"

10. "Obviously" (p 4 line 29) this may not be so obvious, given that microbiota can have systemic effects on e.g. inflammation, which, in-turn, can have effects on a remote cancer, as a "tertiary interaction".

Thank you for bringing this to our attention. We removed "obviously," the sentence was meant to refer to the traditional line of thinking in cancer-microbial relationships.

11. Reference #62 is a set of search terms against the clinicaltrials.gov database, which will be dynamic change over time. Please replace this reference with some sort of a static link, or a table that lists relevant ongoing trials.

Thank you for this point, a table has been included.

12. The discussion of pros & cons of different screening methodologies for colorectal cancer at the top of page 7 does not contribute much. The point that microbiome profiling might improve colon-cancer screening or risk-assessment can be made without wading into this controversial clinical area.

Point taken, the specific screening methods were removed, instead, we focus on microbial-screening specific topics.

Minor points:

1. P5 line 20 " It also facilitates..." can be read as if the bacterium is generating anaerobic conditions. But do you mean the tumor generates anaerobic conditions?

A few colloquial phrases could be made more precise/scientific:

2. "our bodies" (p 1, line 33)
3. " these show up in the literature as" (p 4, line 23)
4. "into your own analyses" (p11, line 47)
5. " that also look at" (legend to Fig 1)
6. "chemo" in Figure 2
7. Typo p5 line 45 " tumore's"
8. P1 line 57 "suggest" not suggests

We have clarified/corrected the suggested phrases and passages above.

Reviewer #2

Topic: The authors describe recent developments in the cancer ~ microbiome field. I agree with the authors that this topic is a worthwhile one which should be of interest to a relatively large audience.

Major Points:

1. The strength of this paper is its discussion of the biological basis of interaction for cancer and the microbiome. I found the text written by the authors to be engaging, detailed, and informative. However, the topic which I found to be lacking was the discussion of the "genome biology" of this interaction, and the findings of the field generated with genomic methods. There is a good discussion of the underlying methods for 16S, and a bit of WGS, but very little review of the results of these studies and whether there are any common threads which add to our understanding of this biological system. I would prompt the authors to revise this manuscript in such a way that it will answer the question: "Have genomic methods of characterizing the microbiome substantively added to our understanding of the interaction between microbes and the development or treatment of cancer?" The authors clearly have a deep understanding of this field, and I think they would be well positioned to answer such a question.

This is a great suggestion to provide some overarching context to the work. We have extensively revised the work and included relevant examples of where there are challenges in the field related to (1) the fact that none of this work would be possible without genomic approaches, (2) where the newness and interdisciplinarity of the field has caused issues, and (3) examples of work being done by researchers and publishers to address these issues moving forward.

Minor Points:

1. It strikes me as odd to start the Introduction with a discussion of the semantic differences of describing the microbiota and the microbiome as physical organisms versus genomic content. I would strongly suggest that the authors reconsider how they would like to best use the unexhausted attention span which the readers will bring to the first paragraph of the manuscript.

This is useful feedback - we have trimmed out excess text in the interest of "getting to the point."

2. There are some odd spacing errors in this text. Please check for " . ", for example on line 13.

We have found and corrected several typos in the text.

Second round
Reviewer 1

The authors have been very responsive to my suggestions. This revision is much stronger and will be of interest to the readers of Genome Biology.

Minor comments

1. "bug" is colloquial (p 2 line 54)
2. Transformation-competent viruses needs hyphen (p3 line 4)
3. p. 7 line 44 - irinotecan is usually referred to without the "hydrochloride" unless one is describing a specific chemical reagent used in an experiment. When describing the clinical drug, omit the salt.
4. in several places the author use 'murine' as a synonym for mouse. This phrase should be reserved for situations in which more than 1 rodent species were studied. If the studies described were all mouse, be specific and say mouse.

Reviewer 2

The authors present a well-written summary and overview of cancer-microbiome research, even going so far as to include a discussion of the use of bacteria as anti-cancer therapeutics. I believe that a newcomer to this field would be able to read this paper and gain a useful understanding of the current state of the field and some of the most important papers to follow in more detail.

Authors' response to reviewers

Reviewer #1

Minor comments 1. "bug" is colloquial (p 2 line 54) - *corrected*

2. Transformation-competent viruses needs hyphen (p3 line 4) – *a hyphen has been added*

3. p. 7 line 44 – irinotecan is usually referred to without the “hydrochloride” unless one is describing a specific chemical reagent used in an experiment. When describing the clinical drug, omit the salt. – *the salt has been removed.*

4. in several places the author use ‘murine’ as a synonym for mouse. This phrase should be reserved for situations in which more than 1 rodent species were studied. If the studies described were all mouse, be specific and say mouse. – *murine has been replaced with mouse, where applicable.*